# Wind Energy Assessment for Renewable Energy Communities

**Sandeep Araveti** [1,2], **Cristian Aguayo Quintana** [3], **Evita Kairisa** [4], **Anna Mutule** [4], **Juan Pablo Sepulveda Adriazola** [5], **Conor Sweeney** [2,6] **and Paula Carroll** [1,2,*]

1  School of Business, University College Dublin, D04 V1W8 Dublin, Ireland; sandeep.araveti@ucd.ie
2  UCD Energy Institute, University College Dublin, D04 V1W8 Dublin, Ireland; conor.sweeney@ucd.ie
3  Department of Computer Science, Université Libre Bruxelles, 1050 Bruxelles, Belgium; cristian.aguayo.quintana@ulb.be
4  Institute of Physical Energetics, LV-1006 Riga, Latvia; evitakairisa97@gmail.com (E.K.); amutule@edi.lv (A.M.)
5  Integrated Optimization with Complex Structure Centre, INRIA, 59650 Villeneuve d'Ascq, France; juan-pablo.sepulveda-adriazola@inria.fr
6  School of Mathematics and Statistics, University College Dublin, D04 V1W8 Dublin, Ireland
*  Correspondence: paula.carroll@ucd.ie

**Abstract:** Renewable and local energy communities are viewed as a key component to the success of the energy transition. In this paper, we estimate wind power potential for such communities. Acquiring the most accurate weather data is important to support decision-making. We identify the most reliable publicly available wind speed data and demonstrate a case study for typical energy community scenarios such as a single commercial turbine at coastal and inland locations in Ireland. We describe our assessment methodology to evaluate the quality of the wind source data by comparing it with meteorological observations. We make recommendations on which publicly available wind data sources, such as reanalysis data sources (MERRA-2, ERA-5), PVGIS, and NEWA are best suited to support Renewable Energy Communities interested in exploring the possibilities of renewable wind energy. ERA5 is deemed to be the most suitable wind data source for these locations, while an anomaly is noted in the NEWA data.

**Keywords:** renewable energy communities; renewable energy; wind speed; reanalysis data

## 1. Introduction

EU legislation is moving at pace to ensure a clean and fair energy transition at all levels of the economy. The final Clean Energy Package makes the EU's electricity market more interconnected, flexible, and consumer-centred [1]. The Clean Energy Package measures aim to make different actors in the energy field more competitive and innovative. Member states must respond and transpose the directives into national law to ensure this happens. The Electricity and Renewables Directives are transposed into Irish law to allow Renewable Energy Communities to participate in the electricity market. Similarly, other jurisdictions are considering what legal, regulatory, grid, and market codes need to be amended to comply with the directives and unlock the potential of energy communities.

The Electricity Directive is consumer-focused and outlines requirements for Member States and Regulatory Authorities to develop frameworks that allow for consumer participation in energy markets [2]. The Directive contains two definitions of an energy community: Citizen Energy Community, which is contained in Directive (EU) 2019/944 (recast Electricity Directive), and Renewable Energy Community, which is contained in Directive (EU) 2018/2001 (the recast Renewable Energy Directive). In both cases, the communities are autonomous legal entities based on open and voluntary participation with the purpose of providing environmental, economic, or social community benefits for its shareholders or members rather than financial profits. Energy communities are entitled to generate, consume, store and sell renewable energy and may be allowed to participate

in cross-border electricity exchanges. In this paper, we use the term "renewable energy community" interchangeably with the term "local energy community".

Energy communities can be realised in various legal forms, which depend on national and regional regulations. For example, Community Benefit Societies (CBS) are a legal structure for community-led initiatives and exist to serve the broader interests of the community. Any profit made by a CBS must be used for the benefit of the community. The appropriate use of the CBSs model in Community Renewable Energy projects is to sell electricity directly to the national grid and reinvest the profits generated for the benefit of the community.

Social Enterprises combine different economic, social, and environmental goals at the core of their activities. Social enterprises are incredibly diverse across Europe, encompassing a range of organisational and legal forms and statuses [3]. These enterprises express their commitment to their social goals by limiting the distribution of surplus income to members and instead reinvesting this for future development.

Cooperatives are community-owned social and economic enterprises. They are the dominant institutional architecture for community renewables in Europe and are gaining popularity. There are a growing number of energy cooperatives in the EU, notably in countries like Belgium, Denmark, Germany, France, and Spain but increasingly also in the other Member States. In Greece, the 2018 law on energy communities adopted cooperatives as the basis for its definition of energy communities. In Sweden, Renewable Energy Communities must take the legal form of an Economic Association, which is their adapted form of a cooperative. There are many other forms and interpretations of energy communities, the review in [4] synthesises 183 definitions across three dimensions of meanings (interpretations of the community concept rather than legal forms or definitions), activities, and objectives of the community. The authors note the foremost meaning ascribed to an energy community is a community associated with a "place", and this is particularly prominent for community wind where locally owned, utility-scale wind development interconnected to the grid is anticipated. We adopt this interpretation that an energy community is expected to conduct projects located in the vicinity of where they live.

There are many challenges to the success of local energy communities (LECs), including policy support, social acceptance, financing, management and market structures, and smart grid technologies [5]. LECs must decide the mix of renewable energy generation and storage, where to locate, and how best to manage the assets to achieve the LEC objectives. Energy communities are not just about technical smart energy systems innovations, but they cannot succeed without technological and decision support. Likewise, the Clean Energy Package and National Energy and Climate Plan (NECP) ambitions of EU Member states cannot be achieved without energy community and energy citizen participation.

In Ireland, the Renewable Electricity Support Scheme (RESS) supports community social enterprise and community participation in the electricity market through a competitive auction-based framework [6]. Through a community enabling framework supported by the Sustainable Energy Authority of Ireland (SEAI), citizens can form a Renewable Energy Community and then apply for a license at RESS auctions. The auctions are designed to support progress toward the 80% renewable electricity target in Ireland's NECP [7].

Wind energy development usually presents greater challenges in terms of planning and engineering. However, in the case of Ireland, which lies at 53° North and has an extensive coastline and temperate maritime climate, wind projects are likely to provide a higher Community Benefit Fund (CBF) return compared to solar. The CBF is a mandatory contribution of €2/MWh for all generation projects to be used for the wider economic, environmental, social, and cultural well-being of the local community. Figure 1 shows estimated wind speeds at 20 m elevation over Ireland based on measurements taken during 2001–2010 [8].

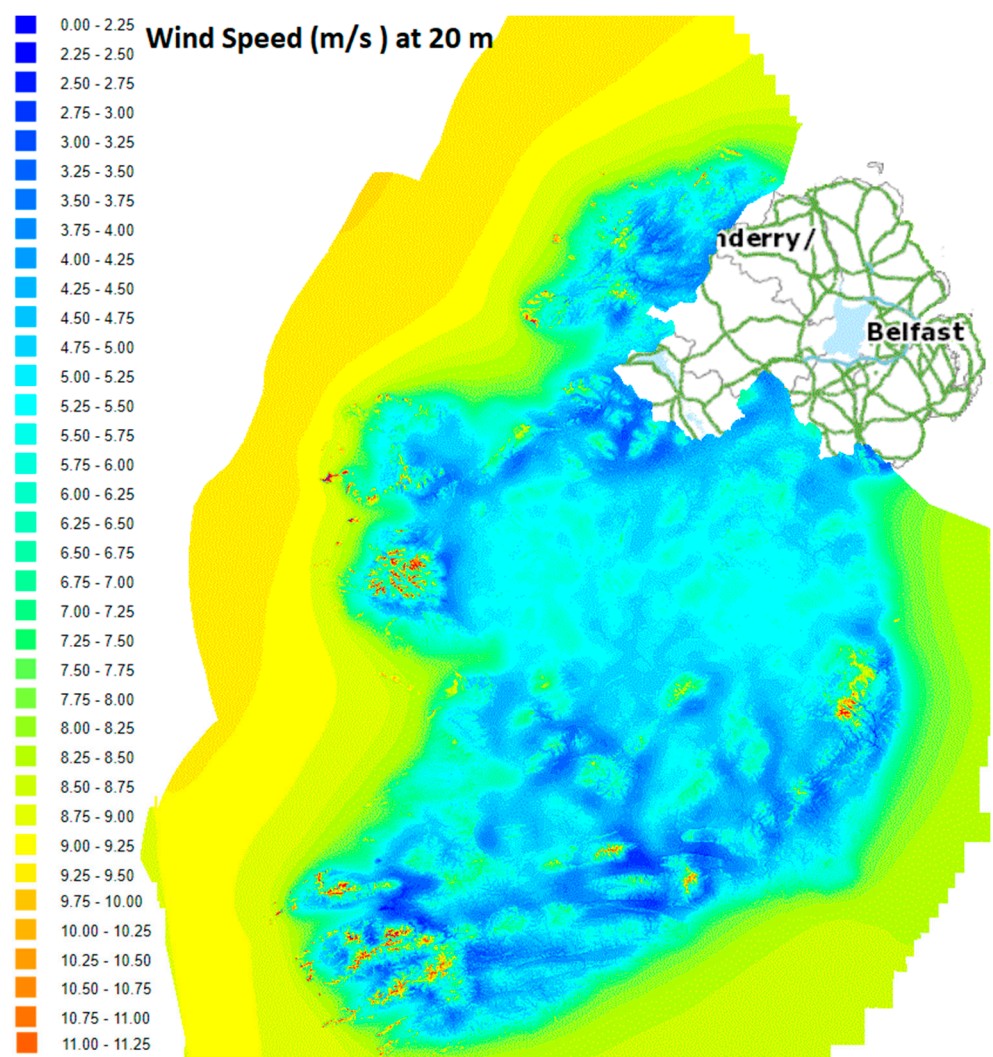

**Figure 1.** Wind Speeds m/s at 20 m elevation over Ireland–, adapted with permission from [8]. 2022, SEAI.

Micro wind generation means a wind turbine or turbines with a nominal output of 300 watts or more but no more than 50 kilowatts with hub heights of 10–15 m. Planning permission is required for the installation of all but the smallest turbines. While lower turbine heights may be allowed in urban settings, the technology to date has not been cost-effective [9]. Very low performances are observed for a micro-wind turbine in an urban area in Italy for most of the year, and the technology is deemed not to be suitable or cost-effective [10]. A possible case for micro wind turbines in agricultural farms is made in [11] but is dependent on a longer than usual payback period of 12 years and state grants. The same team of authors at the University of Southampton outlined the "Boom-to-bust" history of micro wind in [12] and concluded that half of the urban turbines evaluated in their study have a negative load factor, that is, they consume more power than they generate over the year.

The turbines for commercial wind farms are 100 m or higher, where wind speeds are higher than those measured at ground level. Community-led projects in the RESS scheme have an upper limit of 5 MW, effectively restricting the design choice to one or two wind turbines [13]. Wind farm technology is constantly improving, and investment costs are reducing. The estimated costs are around €1.4 million per megawatt installed for the types of wind turbines a LEC would use (1.5 to 3 MW in size) [14]. The wind turbine itself incurs

the most significant costs (~€1.25 M–€2.5 M). Additional costs include groundworks, grid connection, planning, and consultancy costs.

Evidence of long-term wind speed trends is required to secure financial backing for a project. Wind speed is the most significant factor in determining the amount of wind power that can be produced, but wind speeds are difficult to predict and highly variable. The wind at coastal locations is strongly influenced by changes in sea-surface temperatures (breeze circulations), friction, as well as local topography. Furthermore, wind resources at a specific site are difficult to quantify with any level of confidence without undertaking field trials. Potential LECs are advised to first utilise the best available wind speed data and estimation tools and then install anemometry to determine the observed wind speed distribution at a particular site.

Reanalysis provides a picture of past weather and climate. Reanalysis blends observations with past short-range weather forecasts rerun with modern weather forecasting models. Reanalysis data is produced by running a numerical weather prediction (NWP) model in hindcast mode, i.e., a projection back in time rather than a forecast prediction of future values. The reanalysis models assimilate historical observations and produce a time series of historical weather variable values. Three reanalysis data sets are evaluated in [15]. The authors conclude that the choice of dataset depends on the type of renewable energy and the location; there may not be a single best source for wind and solar at all locations.

The authors in [16] note that ideally, wind speed should be measured at a typical turbine hub height. However, there are no long-term records of hub height wind speed available at different locations around Ireland, and anemometry equipment can be expensive. Therefore, their study focuses on 10 m wind speed observation records from synoptic stations, using extrapolation to estimate the wind speed at 100 m hub height.

Reliable historical wind speed and solar irradiance databases are fundamental for policymakers and the distribution system operator (DSO). Rapid variations in wind and solar power availability may compromise the secure operation of the grid. The intermittent nature of renewable energy can result in voltage fluctuations and a decrease in power quality. Even with the high penetration of renewable energy, the low voltage (LV) network must be able to keep the voltage levels and system frequency within the allowable limits while the balance between power generation and demand remains stable. Renewable energy scenarios provide a framework for exploring future energy perspectives, give insight into the behaviour of complex systems, and demonstrate both benefits and challenges associated with the increased integration of renewables into the LV network.

Additionally, data-driven management strategies will determine the short-term decision-making of prosumers in local energy markets. The design of markets and policies such as grid usage pricing heavily depends on modelling prosumers' behaviour [17,18]. The design of such policies as a function of minute-based operation may bring opportunities for coping with the rapid variations of the uncertainty. The data used to feed those models should capture the complex spatiotemporal correlations found in actual scenarios [19]. A higher sampling frequency (in the order of the seconds) is required to perform out-of-sample simulations of real-time operations.

With regards to energy planning, the problem of unit commitment is historically one of the most important tools in power systems. This is a challenging optimisation problem in planning which thermal and hydro generation units to "commit" to serve the electricity demand. Recently, renewable energies have been incorporated. The conventional generation is only required to serve the net demand. The problem becomes more difficult to solve when variable wind generation is included [20]. Renewable energy scenarios play a fundamental role in obtaining robust planning and production schemes [21].

In this paper, we focus on identifying the most suitable publicly available wind speed data for LECs to evaluate the potential for renewable wind as part of an Initial Viability Study. We demonstrate the approach at a sample set of locations in Ireland. The paper is organised as follows: Section 2 gives a detailed overview of the data and methodology used for the quality assessment. Section 3 presents the results of the quality and accuracy

assessment. Section 4 discusses the insights on identifying the best data source. Finally, conclusions are highlighted in Section 5.

## 2. Materials and Methods

Since micro-wind has been shown to have limited potential, we focus on a scenario of a LEC that wishes to install a single commercial wind turbine. We consider wind speed at a hub height of 110 m for a potential community-owned onshore wind turbine. We first evaluate two important reanalysis datasets, ERA-5 and MERRA-2, one high-resolution mesoscale data (NEWA), and four PVGIS datasets (two satellite and two reanalysis datasets) by comparing them with the Met Éireann hourly wind speed data observations at four locations in Ireland over a common data period 2009 to 2016. The objective of this step is to identify the best wind speed data source for LECs in Ireland.

Once the best data source was identified, we then assessed the energy potential at a sample of 13 locations in Ireland as a case study to demonstrate the methodology. Seven of the 82 licensed projects in the first RESS auction in 2021 are majority-owned by communities, and the revenues from their operation will be cycled back into those communities. Table 1 shows a summary of the five solar energy and two onshore wind community projects that were successful.

**Table 1.** Location of Local Energy Community Projects licensed in the RESS 2020 auction in Ireland.

| Company Name | Project | RES Type | Capacity MW | County |
|---|---|---|---|---|
| CEARTH Ltd.,Kinsale, Co. Cork | Dooleeg More Wind Farm | Onshore Wind | 2.5 | Mayo |
| Clooncon East Single WTG Ltd. Ballinlass, Co. Galway | Clooncon East Single WTG | Onshore Wind | 0.9 | Galway |
| Ballytobin Solar Ltd. Dublin D20 A970 | Ballytobin Solar PV | Solar | 4 | Kilkenny |
| Davidstown Renewables Ltd. Wexford | Davidstown Solar | Solar | 4.95 | Wexford |
| I.Q Solar Ltd. Cloyne, Cork | Lurrig Solar Farm | Solar | 4 | Cork |
| Templederry Energy Resources Ltd. Nenagh Co Tipperary | Barnderg Solar Farm | Solar | 4 | Galway/Mayo |
| Templederry Energy Resources Ltd. Nenagh Co Tipperary | Lisduff Solar Park | Solar | 4 | Clare |

In addition to the types of locations where community projects were licensed, we consider six additional locations taken from [15], based on the availability of continuous records of hourly data. Figure 2 shows the 13 locations in our study that span the island of Ireland geographically. These stations represent the longest available record of wind and shortwave radiation covering different regions of Ireland. The locations include eight coastal locations (Galway, Wexford, Mayo, Clare, Belmullet, Dublin Airport, Malin Head, and Valentia Observatory) and five inland stations (Kilkenny, Cork, Birr, Clones, and Mount Dillon).

In the final step, we extract wind speeds for the sample of 13 locations in Ireland from the best data source. Wind speed is then converted to wind power using a sample manufacturer's turbine power curve.

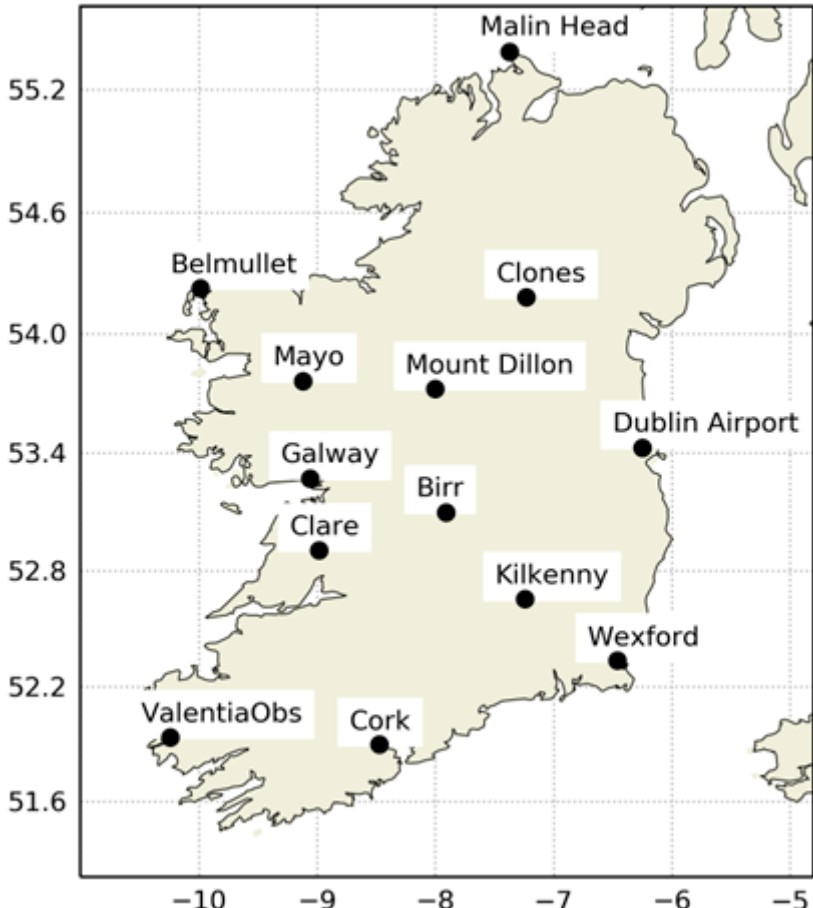

**Figure 2.** Thirteen study locations in Ireland.

### 2.1. Data Sources

The availability of wind speed data at hub height is limited. The Met Éireann "ground truth" wind observations are available at 10 m height only. Therefore, we use 10 m wind speed records for the quality assessment. The global reanalysis (ERA-5 and MERRA-2), satellite datasets (SARAH and CMSAF), regional reanalysis (ERA-5 and COSMO), and station data (Met Éireann) were available at a one-hour temporal resolution, whereas the NEWA data was available at 30 min resolution, all freely available. Table 2 shows a summary of the data sources which were assessed in our study. In the following sections, we describe the data sources in more detail.

**Table 2.** Reference datasets used in the study over a common data period of 2009 to 2016.

| Parameter | ERA-5 | MERRA-2 | NEWA | PVGIS-SARAH | PVGIS-SARAH2 | PVGIS-ERA5 |
|---|---|---|---|---|---|---|
| Model/Satellite | IFS Cycle 4lr2 | GEOS v5.12.4 | WRF v3.8.1 | EUMETSAT | CMSAF | IFS |
| Institution | ECMWF | NASA | NEWA | JRC | JRC | ECMWF |
| Horizontal grid | 0.25° × 0.25° (~31 km) | 0.5° × 0.625° (~70 km) | 0.027° × 0.027° (~3 km) | 0.05° × 0.05° (~5 km) | 0.038° × 0.0.38° (~4 km) | 0.038° × 0.0.38° (~4 km) |
| Time coverage | 1979–2021 | 1980–2021 | 2009–2018 | 2005–2016 | 2005–2020 | 2005–2020 |
| Time resolution | 1 h | 1 h | 30 min | 1 h | 1 h | 1 h |
| Spatial coverage | Global | Global | Europe | Europe, Asia, Africa, and South America | Europe, Africa, parts of South America | Europe |

### 2.2. Met Éireann Observational Data

Met Éireann is the Irish Meteorological service and has an observation network that gathers weather data across the country for use in weather forecasts, aviation, and meteorological research. Their aim is to produce data of the highest quality that is widely

available and very easy to access; they also have researcher-friendly access. The weather observations (for example, temperature, relative humidity, wind speed, wind direction, shortwave radiation, and rainfall) record day-to-day changes in the atmosphere and are quality controlled and archived in the Met Éireann database [22]. This historical data is available at different temporal scales such as hourly, daily, or monthly for various weather variables. The Met Éireann dataset was considered the "ground truth" in this paper against which the reference data were compared to check their quality and accuracy.

### 2.3. ERA-5

ERA-5 is a fifth-generation reanalysis by the European Center for Medium Range Weather Forecasts—ECMWF. Hourly data of many atmospheric, land, and oceanic climate variables such as wind speed at 10 m and 100 m elevations from 1979 to the present at a spatial resolution of 0.25° × 0.25°, i.e., a ~31 km grid [23]. ERA-5 includes information about uncertainties for all variables at reduced spatial and temporal resolutions. The data covers the entire Earth. Studies have shown ERA-5 data fit the ground measurements for Ireland well [15] and have found its usage satisfactory for energy system modelling [24]. ERA-5 can be accessed from the climate data store. The downloading procedure is through the web interface of CDS. A procedure to download the ERA-5 data is outlined in [25].

### 2.4. MERRA-2

The second version of Modern-Era Retrospective analysis for Research and Applications (MERRA-2) data is produced by NASA [26] using an upgraded version of the Goddard Earth Observing System Model version 5.12.4 (GOES-5) based on three-dimensional variational data assimilation system. MERRA-2 is one of the few global reanalyses that assimilate the data from the entire constellation of NASA satellites. MERRA-2 data is available from 1980 to the present day, with global coverage of 0.5° lat × 0.625° lon (horizontal grid resolution). MERRA-2 data can be downloaded from NASA Goddard Earth Sciences (GES) data and information service center. This procedure is outlined in [27].

### 2.5. NEWA

The New European Wind Atlas (NEWA) is a joint research effort from eight European countries and is a combination of downscaling of ERA-5 data and Weather Research Forecast (WRF) model data. It is a comprehensive mesoscale model with a spatial coverage of the whole of Europe and Turkey, extending at least 100 km from any known coastlines. Data are available at 30 min temporal resolution for 2009–2018. It is driven by ERA-5 with special nudging over a domain. Nudging, or Newtonian relaxation, is a simple form of data assimilation that adjusts dynamical variables of free-running Global Climate Models (GCMs) using meteorological reanalysis data to give a realistic representation of the atmosphere at a given time. All mesoscale simulations in NEWA use three nested domains with a 3 km horizontal grid spacing for the innermost grid and a 1:3 ratio between inner and outer domain resolution, leading to three different resolutions: 27 km for the outer domain and 9 and 3 km for the inner nested domains. The model details can be seen in [28]. The temporal coverage of the sensitivity simulations is 1 year (2015 or 2016), based on data availability. WRFs over 10 domains cover most of Europe. NEWA data can be accessed via a web interface that includes interactive maps, time series of wind speed and direction at different heights, as well as other information of relevance to the wind industry. Full details of the NEWA model are described in [29,30].

### 2.6. PVGIS

While the focus of this paper is on wind energy assessment, we mention here sources that provide additional focus on solar irradiance as many LECs will be interested in selecting the best mix of renewable energy sources to meet their goals. In other works, we evaluated the reference data sources with respect to temperature and solar irradiance. PVGIS version 5.2 is a tool developed at the European commission joint research center

that provides meteorological data. The main focus of PVGIS is solar resource assessment and photovoltaic (PV) performance studies, but hourly wind speed at 10 m at high spatial scales is also available. Data is available on the website [31]. PVGIS 5.2 provides three different sub-datasets (a) PVGIS SARAH, (b) PVGIS SARAH2, and (c) PVGIS-ERA5.

### 2.6.1. PVGIS-SARAH

The Surface Solar Radiation Data Set-Heliosat (SARAH) is a satellite-based climatology of the solar surface irradiance, the surface direct normalised irradiance, and the effective cloud albedo derived from satellite observations of the visible channels of the MVIRI and SEVIRI instruments onboard the geostationary Meteosat satellites. The data are available from 2005 to 2016 and cover the region ±65° longitude and ±65° latitude. The products are available as monthly, daily, and hourly averages on a regular latitude/longitude grid with a spatial resolution of 0.05° × 0.05°. For more detailed information, see [32].

### 2.6.2. PVGIS-SARAH2

The Satellite Application Facility on Climate Monitoring (CMSAF) solar radiation product SARAH-2.1 (PVGIS-SARAH2) provides data from 2005 to 2020. The CMSAF solar surface irradiance retrieval is based on radiative transfer calculations using satellite-derived parameters as input. It is a part of the European Organization for the Exploitation of Meteorological Satellites (EUMETSAT) ground segment and part of the EUMETSAT network of Satellite Application Facilities.

### 2.6.3. PVGIS-ERA5

Data from the ERA-5 ECMWF are processed to higher resolutions. The original ERA-5 data is gridded into a higher spatial resolution, 4 by 4 km. The PVGIS ERA5 data covers the Earth over a time frame from 2005 to 2020 and provides 137 elevation levels from the surface up to a height of 80 km.

### 2.7. Methodology for the Quality Assessment

Four skill scores are used to compare the reference data sets in Table 2 to the Met Éireann ground truth measurements. The statistical metrics are; correlation coefficient (*R*), mean error or bias (ME), root-mean-square error (RMSE), and index of agreement (IOA). $x_i$ are the estimates and $y_i$ are the Met Éireann observations. *N* is the number of data points for analysis. $\bar{x}$ and $\bar{y}$ are the average of the estimated and observed values, respectively. We define each metric next.

$$R = \frac{\sum (x_i - \bar{x})(y_i - \bar{y})}{\sqrt{\sum (x_i - \bar{x})^2 (y_i - \bar{y})^2}} \tag{1}$$

Equation (1) is the correlation coefficient. The numerator of Equation (1) is the co-variance of two variables, and the denominator is the product of standard deviations of two variables.

$$\text{Bias} = \frac{\sum_{i=1}^{N}(x_i - y_i)}{N} \tag{2}$$

Equation (2) is the difference between model-predicted and observed values i.e., the mean model error or "bias". Bias is the tendency of a statistic to overestimate or underestimate a parameter. A positive value signifies that the model has overestimated the actual value. A negative value signifies that the model has underestimated the actual value of the period.

Equation (3) is the root mean squared error (RMSE).

$$\text{RMSE} = \sqrt{\frac{1}{N} \sum_{i=1}^{N}(x_i - y_i)^2} \tag{3}$$

Finally, the Index of Agreement (IOA) is a standardised measure of the degree of model prediction error which varies between 0 and 1 [33]. It represents the relative covariability of

the estimates and observations about the 'true' mean. The agreement value of '1' indicates a perfect match, and '0' indicates no agreement at all. The IAO can detect additive and proportional differences in the observed and simulated means and variances; however, IOA is overly sensitive to extreme values due to the squared differences. The IOA is given in Equation (4).

$$IOA = 1 - \frac{\sum_{i=1}^{N}(x_i - y_i)^2}{\sum_{i=1}^{N}(|x_i - \overline{y}| + |y_i - \overline{y}|)^2} \tag{4}$$

*2.8. Wind Power Model*

Wind speed is the most significant variable in estimating wind to electrical energy transformation. The validity of wind power estimates is significantly affected by the quality of the input wind speed data. Wind speed measurements are rarely available to Local Energy Communities without the installation of anemometry equipment and present a challenge to long-term strategic decisions. Hence we recommend the use of sources such as those in Table 2 for the initial evaluation of site suitability.

We follow the approach in [34] to generate wind power energy for a particular type or set of turbines. The steps to transform wind speed into wind power are:

1.   Acquire wind speed records for at least two heights;
2.   Extrapolate the wind speeds to the turbine hub height;
3.   Convert hub height wind speed to power output using the manufacturers' power curve.

A common approach is to extrapolate the vertical wind speeds (*WS*) at 10 and 100 m to hub height (*HH*). $\frac{dWS}{dz}$ is the rate of change of *WS* with height $z$ between 10 m and 100 m heights in Equation (5).

$$\frac{dWS}{dz} = \frac{WS_{100} - WS_{10}}{log(100) - log(10)} \tag{5}$$

Equation (6) allows us to estimate the wind speed at hub height, $WS_{HH}$.

$$WS_{HH} = WS_{10} + \frac{dWS}{dz}(log(HH) - log(10)) \tag{6}$$

Once the wind speed records at hub height are derived from Equations (5) and (6), the wind speed time-series values are converted into wind power energy using the appropriate power curve for the selected turbine.

Power curves depict the relationship between wind speed and power and the range of operation of the turbine [35]. The system of equations in Equation (7) shows the relationship. No power is produced when the wind speed is below a cut-in threshold, $V_c$. Power is produced rated according to a polynomial function $P_n$ up to the rated speed $V_r$. The turbine operates at full capacity up to the furling speed $V_f$—the speed at which the turbine cuts out to protect the blades in strong winds. The system is normalised by the capacity factor, the average power output divided by the rated power of the turbine. Hence full production is shown with a capacity of 1.

$$
\begin{aligned}
P(WS) &= 0 & WS < V_c \\
P(WS) &= P_n(WS) & V_c \leq WS \leq V_r \\
P(WS) &= 1 & V_r < WS \leq V_f \\
P(WS) &= 0 & WS > V_f
\end{aligned} \tag{7}
$$

Several power curves for commercial turbines are available from renewable ninja [36]. In this paper, we have chosen a Vestas V110 2.0 MW IEC IIIA turbine as it is suitable for low-wind sites with a cut in wind speed of 3.0 m/s. It has a capacity of 2 MW, in the range suitable for the RESS scheme in Ireland. The hub height is 110 m. Technical information is available at [37].

The power curve from "Ninja Renewables" is in the form of a discrete series of power values versus wind speeds between 0 and 40 m/s with a resolution of 0.01 m/s. Figure 3

shows the "Wind Turbine Power Curves~5 (0.01 ms with 0.00 w smoother).csv" data for the Vestas.V110.2000 turbine. Then, a smoothed power curve data "Wind Turbine Power Curves~5 (0.01 ms with 0.40 w smoother).csv" is considered. Next, we fit a polynomial of sixth order to the discrete power curve data for the range $V_c \leq WS \leq V_r$ to derive the coefficients of a polynomial $P_n$ as given in Equation (8).

$$P_n(WS) = \sum_{i=1}^{i=6} a_i \, WS^i \qquad (8)$$

where $a_i$ is the coefficient of the best-fitted polynomial.

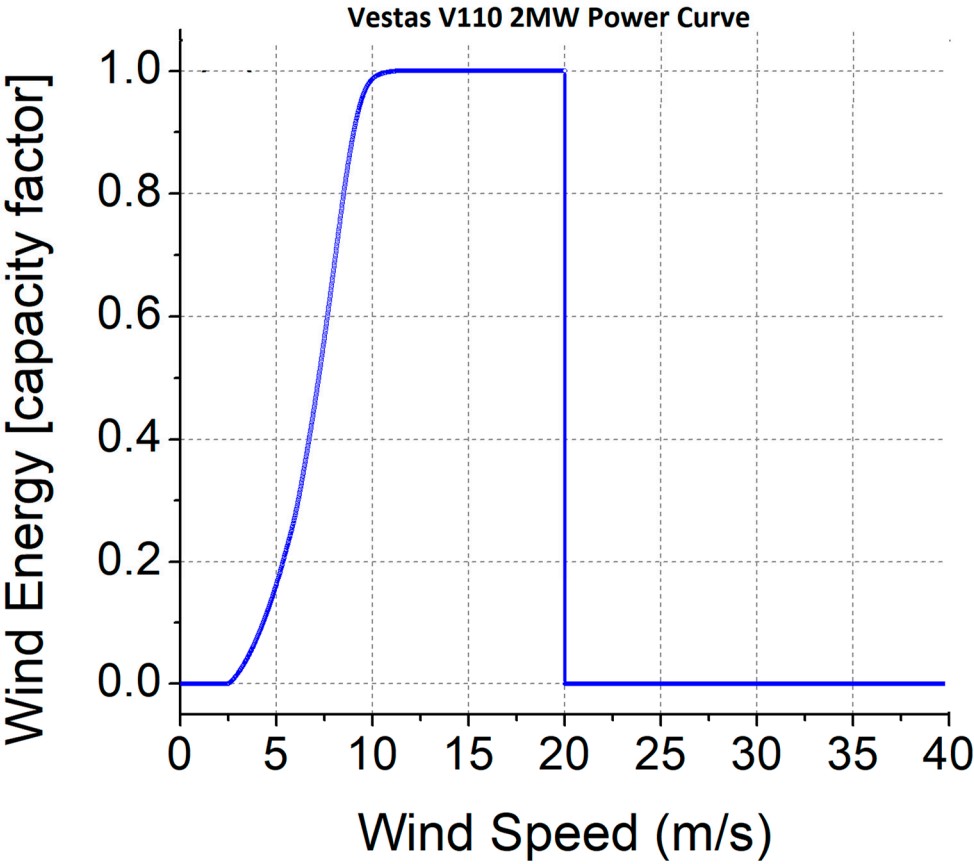

**Figure 3.** The *x*-axis is wind speed. The *y*-axis is the capacity factor for a representative wind power curve (Vestas V110 2.0 MW IEC IIIA).

Table 3 shows the parameters and technical details of the turbine used in our study. $P_n(WS)$ is the normalised power output value. By considering Equations (7) and (8), a time series of WS at hub height is converted to wind power energy. Both the capacity factor and power in Watts are computed at one-hour temporal resolution. The capacity factor (*CF*) is defined as the ratio of average power, *Pavg*, and rated power, $P_r$, i.e., *CF* = *Pavg*/$P_r$.

**Table 3.** Vestas V110 2.0 MW technical details.

| Parameter | Value |
|---|---|
| Cut in speed of turbine ($V_c$) | 3.0 m/s |
| Rated speed of turbine ($V_r$) | 11.5 m/s |
| Furling speed of turbine ($V_f$) | 20 m/s |
| Rated power of turbine ($P_r$) | 2000 kW |
| Wind class | IEC IIIA |
| $P_n$ polynomial coefficients ($a_0, a_1, a_2, a_3, a_4, a_5, a_6$) | $a_0 = 0.03319$; $a_1 = -0.10065$; $a_2 = 0.04862$; $a_3 = -0.00478$; $a_4 = 1.96414 \times 10^{-4}$; $a_5 = -3.71117 \times 10^{-6}$; $a_6 = 2.67285 \times 10^{-8}$ |

We then consider the 110 m wind speeds at the 13 sample locations and generate a corresponding time series of wind energy capacity factors and power (Watts) using the methodology described here to support a LEC in the initial site evaluation.

## 3. Results

In this section, we present the results of our work. We first compare the reanalysis, mesoscale, and satellite datasets listed in Table 2 with the Met Éireann observations over a common data period at four assessment sites. Based on the assessment, we select the best data source for wind speed for wind resource evaluation by a LEC. We then analyse the wind power energy at the 13 sample locations shown in Figure 2.

### 3.1. Quality Assessment of Reference Datasets

The four locations for the quality assessment were chosen because they provide the largest amount of Met Éireann hourly resolution wind records at 10 m height observations from 2009 to 2016. The four meteorological stations for the quality assessment are Dublin Airport, Valentia Observatory, Belmullet, and Mount Dillon. The evaluation metrics are set out in Section 2.7.

Table 4 shows the skill scores of the hourly 10 m wind speed analysis averaged over the four assessment stations. The weaker winds in the ERA-5 reanalysis data were attributed largely to the misrepresentation of surface friction and the elevation differences. At all stations, the diurnal variability was less evident in the three PVGIS datasets, where the wind speeds do not vary ±10% from the mean.

**Table 4.** The average values of skill scores of different reference datasets for the wind speed at 10 m height. The mean is taken over four stations.

| Reference Data | Corr. | Bias | RMSE | IOA |
|---|---|---|---|---|
| Obs vs. ERA5 | 0.89 | 0.23 | 1.47 | 0.92 |
| Obs vs. MERRA2 | 0.87 | 1.28 | 2.06 | 0.88 |
| Obs vs. NEWA | 0.84 | 1.89 | 2.76 | 0.82 |
| Obs vs. PVGIS SARAH | 0.89 | −0.19 | 1.63 | 0.89 |
| Obs vs. PVGIS SARAH2 | 0.89 | −0.19 | 1.63 | 0.89 |
| Obs vs. PVGIS ERA5 | 0.89 | −0.19 | 1.63 | 0.89 |

Overall, the ERA-5 exhibited the lowest value of RMSE and the highest value of correlation and IOA. All the correlations were ≥0.84, suggesting that the referenced datasets are able to capture the observed wind speed at the four stations. ERA-5 and PVGIS showed a higher correlation, followed by MERRA-2. The lowest correlation was seen in the NEWA data. The IOA suggests a similar story with higher values in ERA-5 data (0.92) and lower values in NEWA data (0.82).

As seen in Table 4, positive biases are evident in ERA-5, MERRA-2, and NEWA, and a negative bias is seen in the three PVGIS datasets. The lowest values of bias are recorded in the PVGIS dataset, followed by ERA-5 data. The good performance of ERA-5 may be due

to the newer method of data assimilation (4D-var) in the model compared with the other reanalysis 3D-var version. ERA5 also has a greater number of vertical levels, with the first vertical level closest to the observations at 10 m.

The poorest skill scores were noticed in the NEWA data with higher values of bias and RMSE and lowest values of correlation and IOA. On further investigation, we also observed an issue with the NEWA wind speeds at higher elevations. Figure 4 shows the NEWA wind speeds at 10, 100, and 110 m by season at Dublin Airport as an example. The dip in midday values in Figure 4a could be explained by the mixing of convection currents. However, the coincidence of 10 m, 100, and 110 m wind speeds at midday in Figure 4b,c was unexpected. We would expect to see a difference in the values of wind speed at 10 m compared to 100 or 110 m. This issue was observed at the four assessment locations over the time frame and requires further investigation.

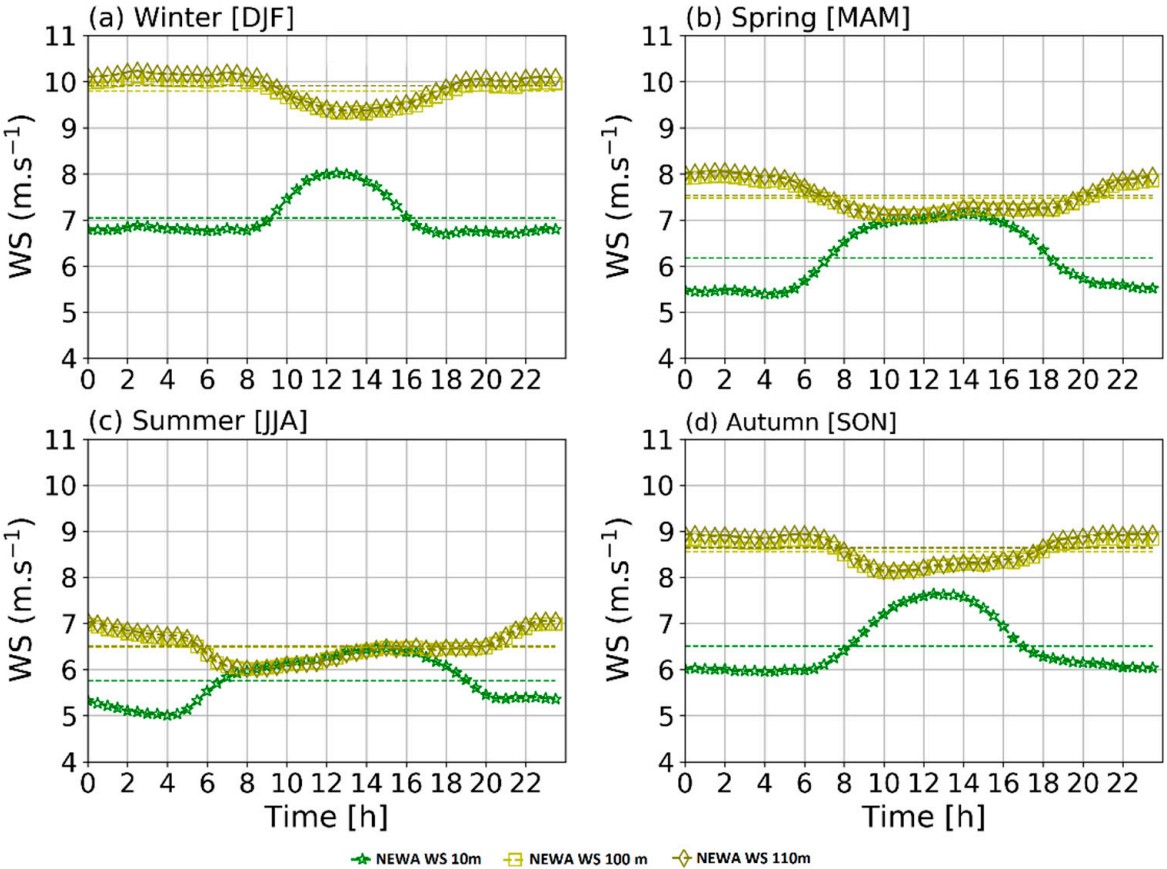

**Figure 4.** NEWA Wind speeds at 10, 100, and 110 m by season (**a**–**d**) and time of day at Dublin Airport over 2009–2016. Note the anomaly with the 100 and 110 m data.

*3.2. Diurnal Variation of Wind Speeds*

Hourly mean wind speeds at the four stations over the time frame 2009 to 2016 are presented in Figure 5. The diurnal cycle (daily pattern) of mean wind speed is of interest in LEC decision making, as historically, electricity demand is higher during the day. The wind experienced at any given location is highly dependent on local topography and other factors. All the stations exhibit significant diurnal variation in wind speed. Stronger winds are evident around noon hours (12–14 IST) compared to the rest of the hours in the observations and all reference datasets. This rise is explained by increased convection currents associated with the rise in temperate during the day.

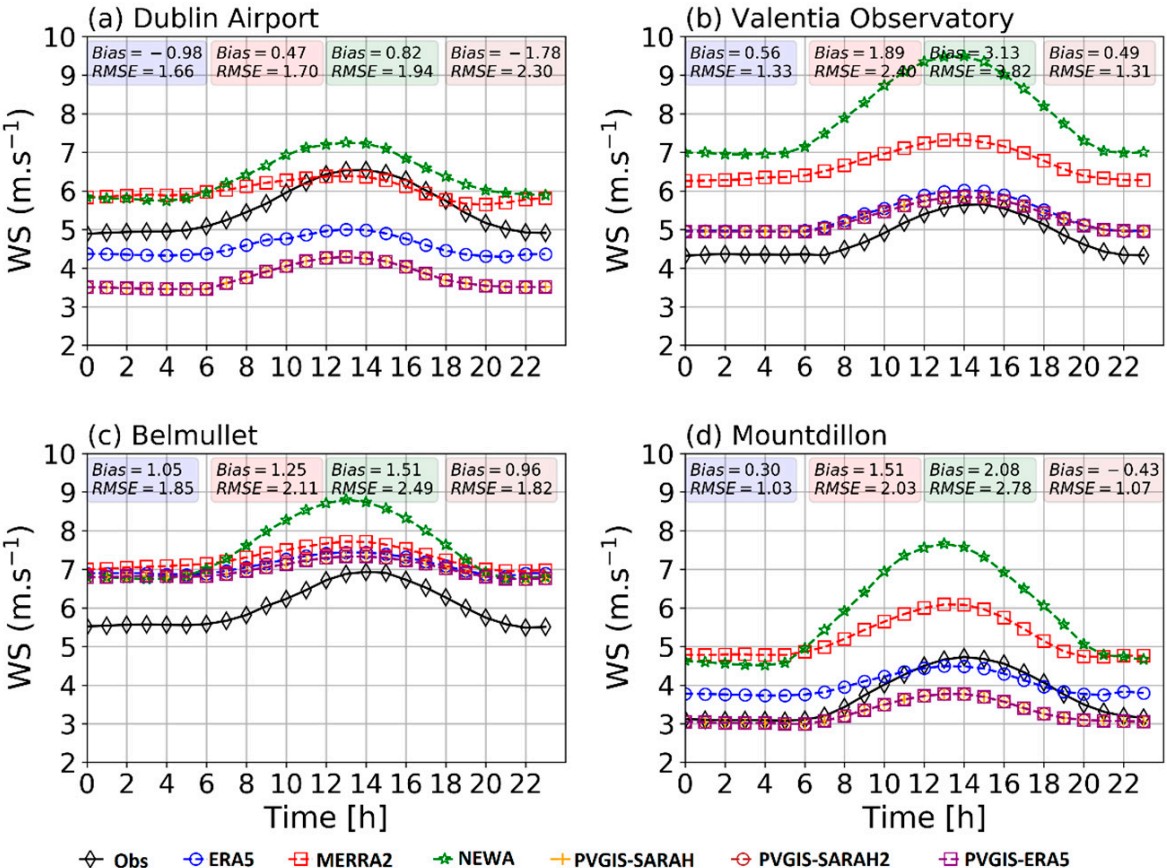

**Figure 5.** Mean 10 m wind speed records at the four assessment locations: (**a**) Dublin Airport, (**b**) Valentia Observatory, (**c**) Belmullet, and (**d**) Mount Dillon stations showing diurnal variation. Black lines represent Met Éireann observations, blue is ERA-5 data, red is MERRA-2 data, green is NEWA data, orange is PVGIS SARAH data, brown is PVGIS SARAH2 data, and purple is PVGIS ERA5 data. The bias and RMSE values between reference datasets and observations are given in the respective colored boxes.

The tendency for maximum wind speed to occur in the afternoon is noticeable only in the long-term figures, such as the time frame in our study from 2009–2016. The maximum may occur at any hour on any individual day.

The diurnal pattern is a surface-driven phenomenon attributed mainly to the oscillation of the transfer of momentum, i.e., the potential for upper-level winds to mix down to the surface. The daily maximum occurring at midday is mostly because of the increased downward transfer of momentum as a result of increased thermal mixing, while during the night-time, when radiative cooling dominates, a stable boundary layer is formed that prevents the transfer of momentum, thus leading to relatively calm winds at the near-surface level [38].

Relatively higher wind speeds were noticed over coastal stations (Figure 4b,c) compared to inland stations (Figure 4a,d). At Mount Dillon, an inland station, the observations show a peak in the wind speed at 14.00 IST with a magnitude of 4.8 m/s, whereas the reference datasets identify a peak one hour earlier with higher wind speeds in NEWA (7.6 m/s) and MERRA-2 (6 m/s) data sets, and lower wind speeds in ERA-5 (4.2 m/s) and all 3-PVGIS datasets (3.1 m/s).

### 3.3. Seasonal Variation of Wind Speed

Wind speed is subject to seasonality. We next explore the diurnal cycle of mean wind speed at 10 m height as a function of different seasons. For climatological and

meteorological purposes, based on air temperature, seasons in the northern hemisphere are regarded as 3-month periods as follows: December to February—winter, March to May—spring, June to August—summer, and September to November—autumn. Results are presented in Figure 6 using Dublin Airport as an example which shows that the diurnal pattern of wind speed changes with the season.

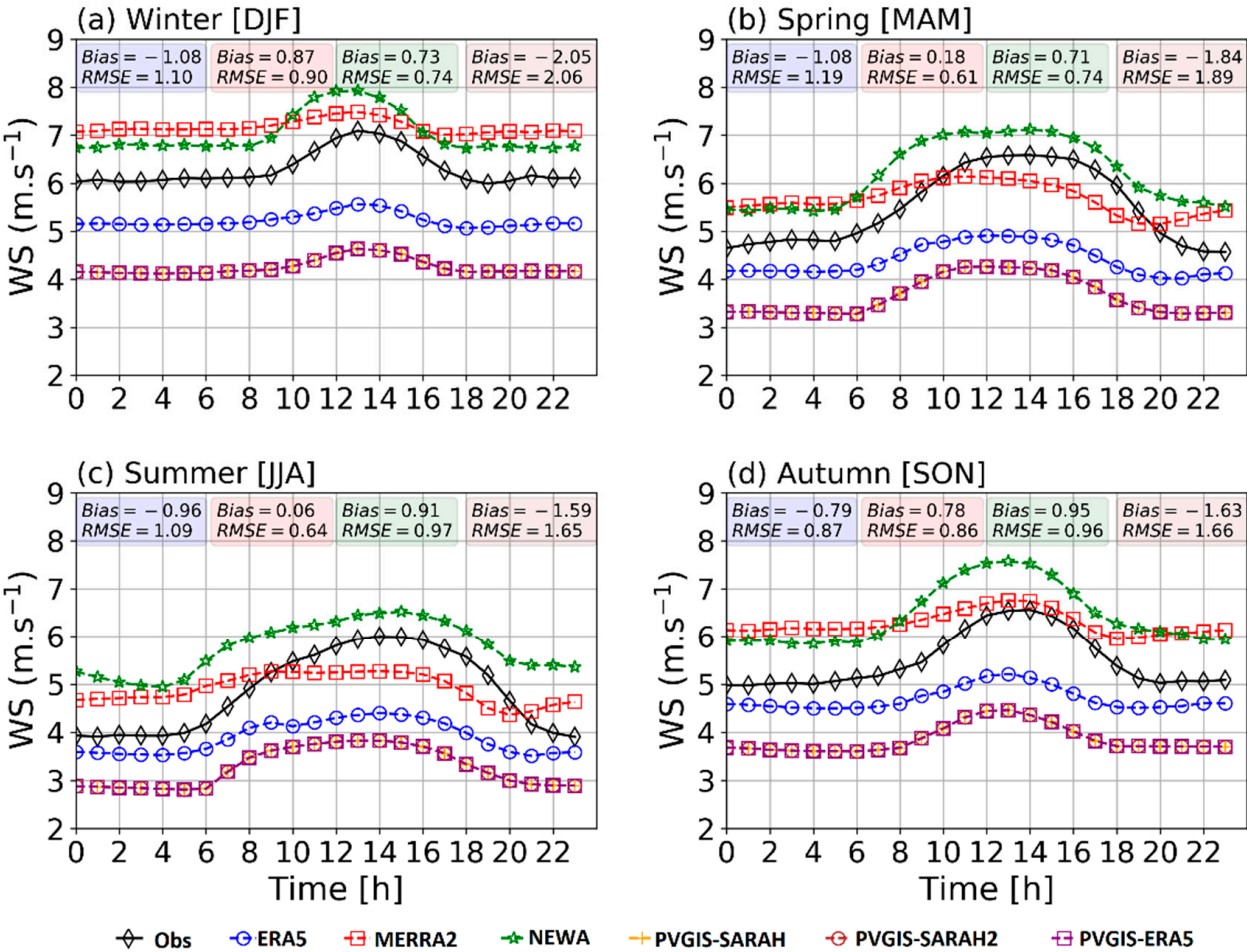

**Figure 6.** Seasonal, diurnal variation of 10 m wind speed at Dublin Airport during the seasons (**a**) winter: December, January, February (DJF); (**b**) spring: March, April, May (MAM); (**c**) summer: June, July, August (JJA); (**d**) autumn: September, October, November (SON).

Similar results were observed for all four assessment locations. Note that Ireland uses daylight savings and experiences short winter days and long summer days. The wind speeds were higher during the midday hours 12.00–14.00 in all seasons. The diurnal variation was less pronounced in summer than in winter. This is a result of surface heating, which increases the mixing of the faster-moving air at higher levels with the air near the surface. As the effect of surface heating diminishes, the wind speed decreases, and during the night, there was little variation from hour to hour.

ERA-5 and the three PVGIS datasets show negative biases at Dublin Airport; the models underestimate the observed wind speeds. In contrast, the NEWA and MERRA-2 data sets show positive biases regardless of the season. Interestingly, the biases were largest during the midday hours 12.00–14.00 compared to other hours.

The correlation and IOQ between observations and reference dataset as a function of the season are shown in Table 5, using Dublin Airport as an example.

Table 5 shows that the minimum correlation for Dublin Airport was 0.8, which indicates that the reference datasets are able to capture approximately 80% variation of the

observed wind speeds. Similar results were observed at all four assessment locations. Higher correlation values occur for the ERA-5 and PVGIS datasets, followed by MERRA-2 data, with the weakest correlation noticed with the NEWA data, irrespective of the season. Interestingly, cooler seasons (winter and autumn) have 5%–10% higher correlation compared to the warmer seasons (spring and summer). This striking feature is clearly evident in all reference datasets.

**Table 5.** Correlations, R, and IOA between Dublin Airport 2009 to 2016 hourly wind speed observations and reference datasets by season.

| Reference Data | Winter R \| IOA | Spring R \| IOA | Summer R \| IOA | Autumn R \| IOA |
|---|---|---|---|---|
| Obs vs. ERA5 | 0.91 \| 0.94 | 0.88 \| 0.93 | 0.83 \| 0.91 | 0.89 \| 0.92 |
| Obs vs. MERRA2 | 0.88 \| 0.82 | 0.86 \| 0.86 | 0.83 \| 0.80 | 0.87 \| 0.81 |
| Obs vs. NEWA | 0.88 \| 0.82 | 0.84 \| 0.76 | 0.80 \| 0.67 | 0.82 \| 0.74 |
| Obs vs. PVGIS SARAH | 0.91 \| 0.93 | 0.88 \| 0.88 | 0.84 \| 0.88 | 0.89 \| 0.92 |
| Obs vs. PVGIS SARAH2 | 0.91 \| 0.93 | 0.88 \| 0.88 | 0.84 \| 0.88 | 0.89 \| 0.92 |
| Obs vs. PVGIS ERA5 | 0.91 \| 0.93 | 0.88 \| 0.88 | 0.84 \| 0.88 | 0.89 \| 0.92 |

Though both the PVGIS and ERA-5 datasets had similar correlation values, the biases and RMSE values were lower in ERA-5 data and were evident in all seasons, see Figures 4 and 5. The sample IOA values in Table 5 also suggest that ERA-5 had a higher agreement with the observations regardless of the season. Hence, we have considered ERA-5 to be the best wind data source for LECs in Ireland to conduct an initial wind resource assessment.

On the basis of the quality metrics in Table 4, we considered ERA-5 to be the best data source for the wind speed evaluation by LECs in Ireland.

*3.4. Wind Energy Assessment*

We proceeded with ERA-5 data to estimate and analyse wind power energy over 13 different locations in Ireland. We use the methodology in Section 2.8 to convert wind speed to wind power for each of the 13 locations by taking the hourly wind the ERA-5 wind speeds at 10 m and 100 m heights for the period 1979 to 2021, a total of 43 years.

Figure 7 shows the mean wind speeds at Dublin Airport as an example. We see that the wind speeds vary with elevation. Higher wind speeds were recorded during the winter season, and lower wind speeds were recorded during the summer season. One point worth noting is that regardless of the season, the diurnal variability of wind speed at 100 m and 110 m heights was less pronounced. In contrast, the wind speed at 10 m exhibited significant diurnal variability with a peak at midday.

Next, we checked the wind speed density. Figure 8 shows a histogram of the 110 m wind speeds, again using Dublin as an example. We observe the typical Weibull shape right-skewed distribution. The left tail shaded in grey shows the wind speeds below the Vestas110 m 3 m/s cutting speed. This accounts for 8.7% of the distribution, 91.2% of the time the turbine is capable of producing power, and the speeds exceed the furling speed only 0.1% of the time.

Finally, wind speeds at 110 m heights are used to calculate the wind power energy capacity factors through the Vestas 110 power curve described in Section 2.8. Table 6 shows the wind speed statistics and expected annual power for all 13 locations. Inland location Kilkenny offered the lowest potential with an estimated average annual production of 7499 MWhs. The average annual demand of residential consumers in Ireland is 4200 kW. At that consumption level, the LEC could potentially support 1785 homes. The incorporation of low carbon technologies such as heat pumps, electric vehicles, and PV panels is expected to significantly change both the demand value and time of occurrence.

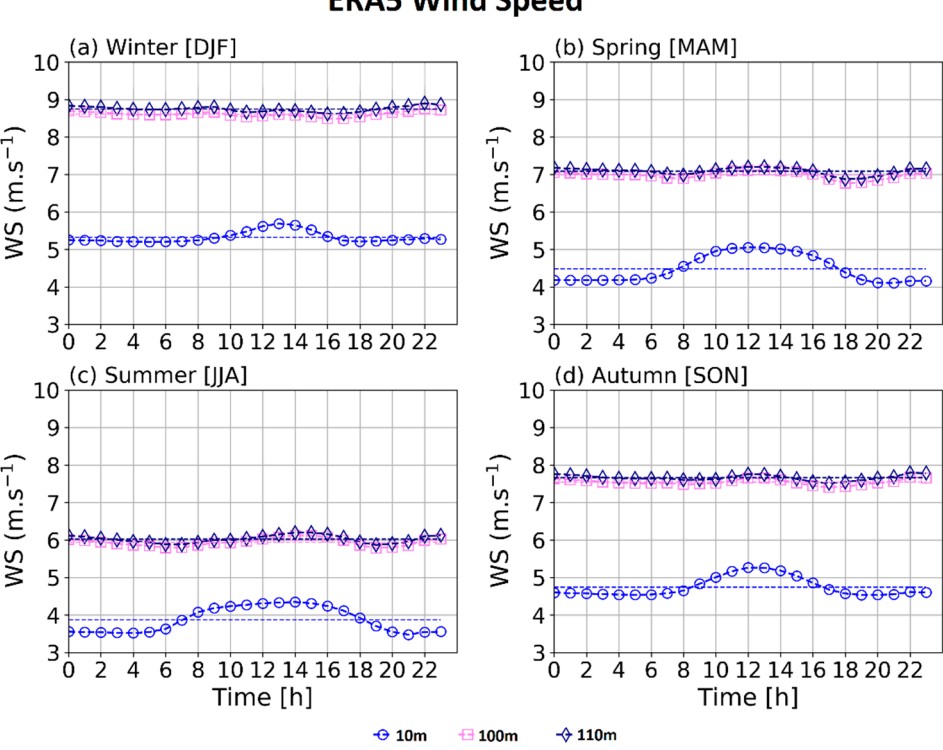

**Figure 7.** Mean ERA-5 wind speed from 1979 to 2021 at different height levels over Dublin Airport during (**a**) winter (DJF), (**b**) spring (MAM), (**c**) summer (JJA), and (**d**) autumn (SON) seasons. Horizontal dashed lines represent the seasonal average wind speed.

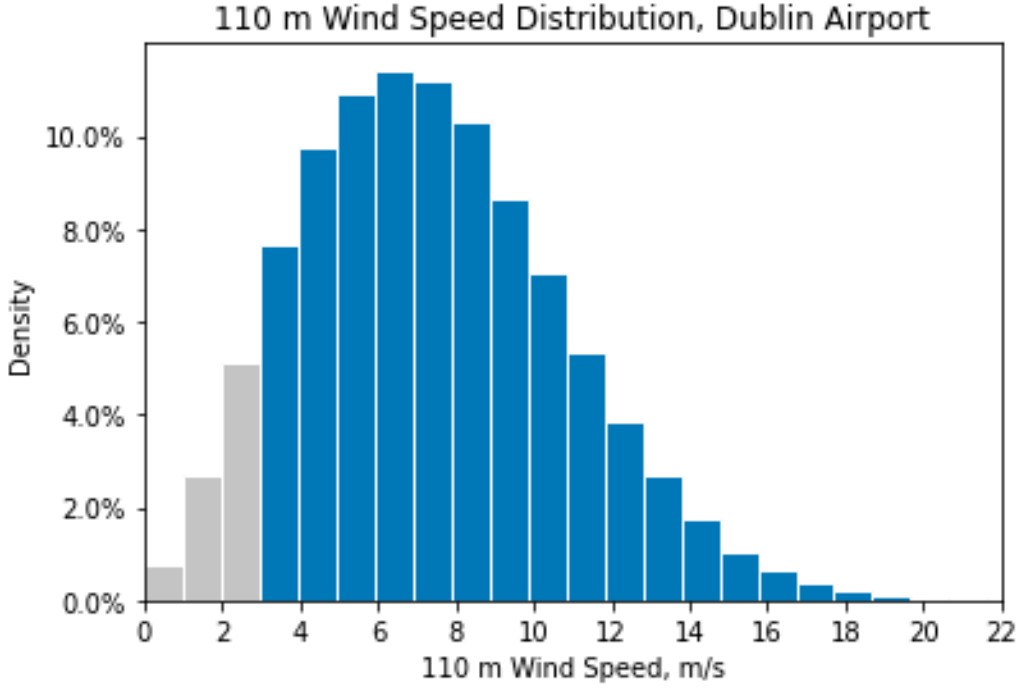

**Figure 8.** ERA-5 wind speed from 1979 to 2021 at Dublin Airport extrapolated to 110 m. Wind speeds below the Vestas cut-in speed of 3 m/s are shown in grey.

**Table 6.** Summary statistics of Wind Speed (m/s) at 110 m hub height, distribution of wind speeds (below cut in $v_c$, within the operating range, and above the furling speed $v_f$), and wind energy production using a Vesta 110 m turbine. The last two rows show the estimated annual electricity production (MW) and the estimated number of average residential consumers that could be served by the turbine.

| WS110 m | Dublin | Belmullet | Birr | Clare | Clones | Cork | Galway | Kilkenny | Malin Head | Mayo | Mount Dillon | Valentia | Wexford |
|---|---|---|---|---|---|---|---|---|---|---|---|---|---|
| Mean | 7.37 | 9.64 | 6.84 | 7.21 | 6.89 | 7.34 | 7.42 | 6.68 | 9.38 | 7.12 | 6.97 | 8.24 | 8.58 |
| Median | 7.09 | 9.17 | 6.60 | 6.89 | 6.64 | 7.00 | 7.03 | 6.43 | 9.02 | 6.79 | 6.71 | 7.82 | 8.16 |
| Mode | 5.98 | 8.33 | 6.26 | 5.79 | 5.96 | 6.04 | 6.67 | 5.85 | 9.18 | 6.28 | 6.33 | 7.55 | 7.01 |
| StDev | 3.39 | 4.57 | 2.98 | 3.28 | 2.98 | 3.34 | 3.37 | 2.96 | 4.43 | 3.17 | 3.01 | 4.04 | 4.06 |
| Kurtosis | 0.07 | 0.16 | 0.33 | 0.56 | 0.30 | 0.51 | 0.64 | 0.37 | 0.07 | 0.47 | 0.41 | 0.19 | 0.35 |
| Skewness | 0.48 | 0.54 | 0.49 | 0.60 | 0.49 | 0.61 | 0.65 | 0.52 | 0.48 | 0.59 | 0.51 | 0.56 | 0.59 |
| Range | 27.55 | 36.26 | 27.68 | 29.61 | 25.85 | 30.06 | 31.27 | 28.41 | 35.46 | 29.64 | 26.67 | 32.80 | 31.24 |
| Below $v_c$ | 8.73% | 5.29% | 8.83% | 8.37% | 8.39% | 8.21% | 7.48% | 9.71% | 5.99% | 7.98% | 8.08% | 8.34% | 6.74% |
| Operating | 91.20% | 92.54% | 91.15% | 91.52% | 91.59% | 91.67% | 92.35% | 90.28% | 92.41% | 91.95% | 91.89% | 91.06% | 92.35% |
| Above $v_f$ | 0.06% | 2.17% | 0.02% | 0.11% | 0.02% | 0.12% | 0.17% | 0.01% | 1.60% | 0.06% | 0.03% | 0.60% | 0.91% |
| Average Annual Production MW | 8532 | 10,883 | 7767 | 8284 | 7843 | 8471 | 8566 | 7499 | 10,756 | 8150 | 7977 | 9525 | 9951 |
| #average homes served | 2031 | 2591 | 1849 | 1972 | 1867 | 2016 | 2039 | 1785 | 2561 | 1940 | 1899 | 2267 | 2369 |

Sample wind power results are shown as a function of the season in Figure 9, again using Dublin Airport as an example.

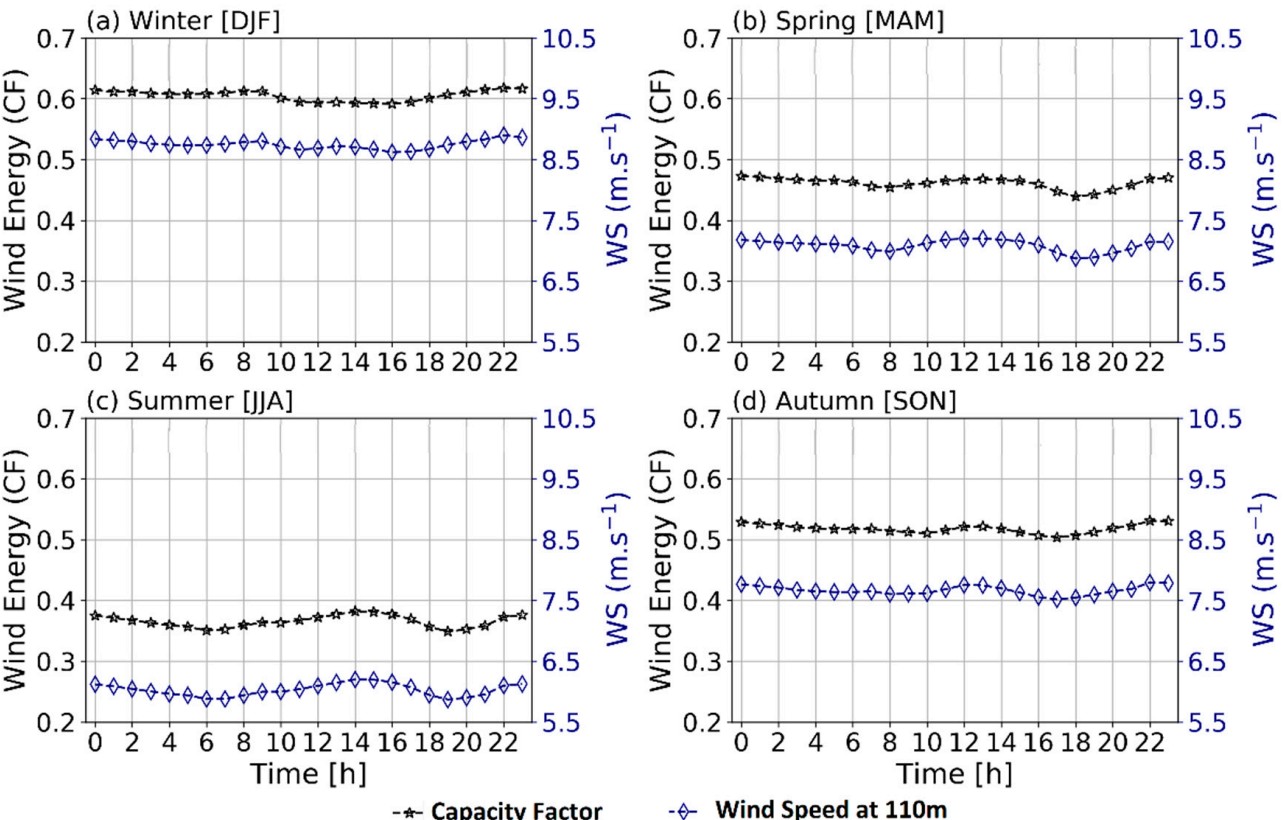

**Figure 9.** Diurnal variation of wind energy at 110 m: capacity factor (left axis in black) and wind speed at 110 m height (right axis in blue) over Dublin Airport during (**a**) winter (DJF), (**b**) spring (MAM), (**c**) summer (JJA), and (**d**) autumn (SON) seasons.

Capacity factors were approximately 10%–15% higher in the cooler seasons (winter and autumn) higher compared to the warmer seasons (spring and summer). On a 24-hour cycle, average wind power dipped during early evening hours (16–18 IST), especially during the summer months. Additionally, a small dip was noticed during morning hours (05–07 IST). However, this pattern did not exist every month. Individual monthly traces of hourly average wind power at other stations also did not have a consistent pattern throughout the year. On average, at Dublin Airport, wind energy was slightly higher around noontime (±2 h). The diurnal pattern is more pronounced at inland stations such as Mount Dillon.

Figure 10 shows the results for the other 12 locations and demonstrates the variation between stations. All stations except Wexford and Malin Head exhibited pronounced diurnal variability in the wind energy capacity factor with a peak at 14 IST, i.e., a time when the maximum amount of energy was generated. Higher capacity factors were recorded in coastal station Belmullet, with the lowest capacity factors estimated in Kilkenny. Capacity factors at the inland stations (Kilkenny, Cork, Birr, Clones, and Mount Dillon) were approximately 10%–20% lower than at coastal stations.

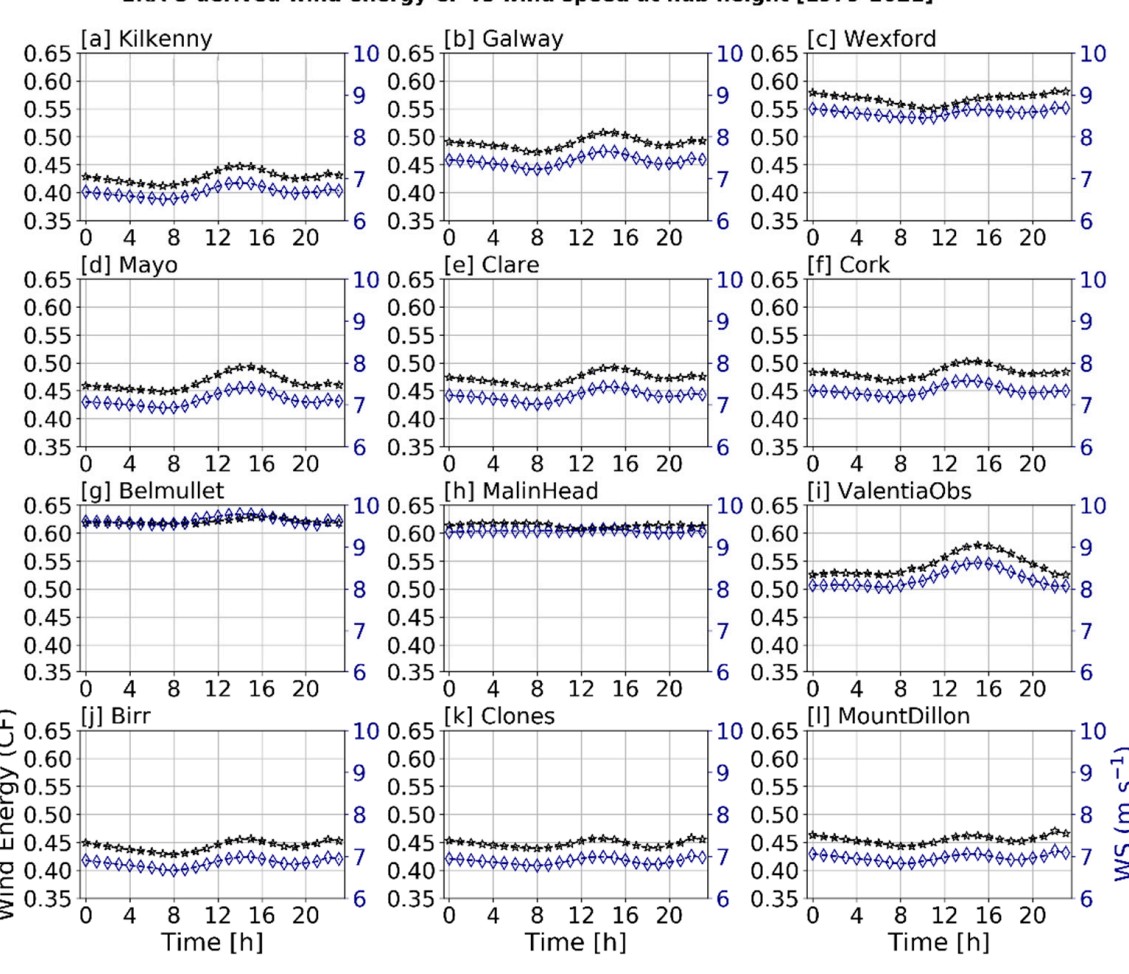

**Figure 10.** Diurnal variation of wind energy capacity factor (left axis) and wind speed at 110 m height (right axis) over 12 different locations as shown in Figure 1 (except Dublin Airport). In all panels, the blue color shows wind speed at 110 m height, and black shows wind energy capacity factors. Subfigures (**a**–**l**) show the 12 locations. The location name is shown on the top left of each subfigure.

## 4. Discussion

This research paper assesses how good the representation of wind speed from various publicly available datasets is in comparison to observations. The objective is to support LECs in their initial evaluation of site suitability. Table 6 shows that across Ireland, a wind turbine such as the Vestas 110 m offers the potential to a LEC to meet the electricity needs of several thousand homes. Wind assessment should be based on long-term time-series data. Short periods consisting of a few years may hide variations from the long-term average and consequently lead to inaccurate wind estimation and poor outcomes for the LEC. Longer periods yield more representative results.

Moreover, understanding the diurnal variations and their predictability is of key importance for the integration and optimal utilisation of wind in the power system. Figure 11 shows representative electricity demand load profiles for residential consumers in Ireland. The profiles are created from a smart meter consumer behaviour trial in Ireland; the data are available from [39]. The electricity load profiles show diurnal patterns with midday peaks that coincide with high wind capacity factors. However, the evening peaks coincide with dips in the wind capacity factors. The wind variation is largely due to the fact that temperature differences, e.g., between the sea surface and the land surface, tend to be larger during the day than at night.

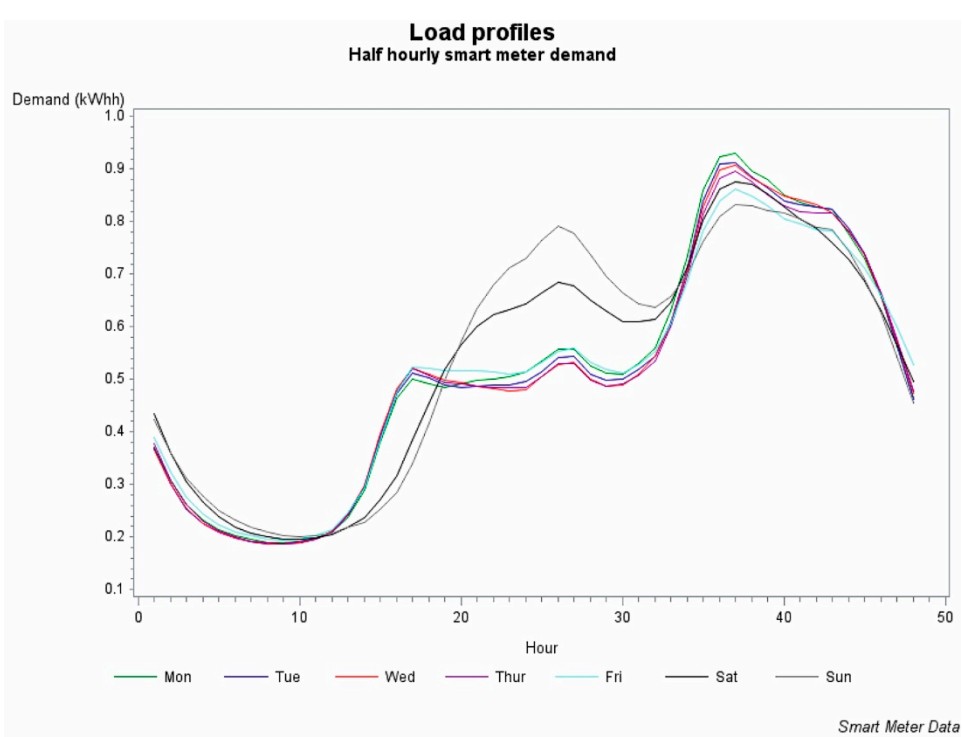

**Figure 11.** Electricity representative load profiles for residential customers in Ireland.

Utilising the long-term measurements (2009 to 2016), we found that ERA-5 outperformed the other sources in Ireland. We found the highest correlation, IOA values, and lowest RMSE values. The choice of dataset to use is important for LECs in their initial wind energy assessment. An overestimation in wind speed during the night often changes to an underestimation in wind speed during daytime hours and vice-versa, see Figures 5 and 6. A false representation of vertical mixing in the lower atmosphere in the models and surface friction, local topography, and many other factors are responsible candidates for these biases. After sunrise, the observed wind speeds increase faster than all other reference datasets. This increase is due to turbulent mixing between different vertical levels, where higher wind speeds are entrained from faster wind speed layers higher up in the atmospheric boundary layer, causing the wind speed at the surface to become faster.

In this paper, we only considered wind-to-power. There are times when electricity demand is low, and wind generators may be instructed to curtail production. Curtailment instructions may also be issued for technical or other operating considerations such as the system's nonsynchronous penetration limit. There are opportunities to explore how wind energy could be stored or converted to other forms to reduce curtailment and improve the cost benefits for wind generators generally or how a renewable energy mix and storage could be combined to serve the LECs needs. As noted, the adoption of low carbon technologies is expected to significantly change both the demand value and time of occurrence. Further research is needed to understand how best to match the available renewable generation with the demand.

We have not explored the difference between the wind power estimates and the actual power produced. The actual performance may differ from that specified by the manufacturer under laboratory test conditions since each site has multiple variables that influence the wind farm performance. There is a gap that needs to be assessed as estimates are often too optimistic and decision support for long-term investments by the energy communities [40].

In future work, we aim to explore additional datasets for the European region and to explore the biases by season, time of day, and particularly by location. Issues in using ERA-5 data at more challenging sites such as mountains, hills, or ridges are noted in [41]

in which it is shown that land-use complexity may negatively affect ERA5 wind energy prediction skills even more so than topography complexity. This will allow us to design more accurate methodologies to support LECs.

## 5. Conclusions

This paper aims to raise awareness of the importance of selecting the most suitable reference dataset. Acquiring the most accurate weather data is important to support the LECs decision-making. All the evaluated reference datasets capture the observed wind speed variation and diurnal and seasonal cycles at the selected four locations in Ireland. The statistical metrics suggest that ERA-5 outperforms other datasets for wind speed. We select ERA-5 as the most suitable publicly available data source for initial site evaluation by LECs in Ireland. However, the biases in the models need further investigation; the analysis of spatial grid behavior differences is beyond the scope of this paper and will be considered for future work

The poorest skill scores and an anomaly at 100 and 110 m wind speeds were noted in NEWA data. The highest resolution dataset may not necessarily be the most accurate. Results also highlight the variability in skill for different locations in Ireland.

**Author Contributions:** Conceptualisation, P.C.; methodology, P.C., C.S. and S.A.; computation, S.A.; validation, P.C., C.S. and S.A.; formal analysis, P.C., C.S. and S.A.; investigation, P.C. and S.A.; data curation, S.A.; writing—original draft preparation, P.C.; writing—review and editing, C.A.Q., J.P.S.A., E.K., A.M., P.C., C.S. and S.A.; visualisation, P.C. and S.A.; supervision, P.C. and C.S.; funding acquisition, P.C.; All authors have read and agreed to the published version of the manuscript.

**Funding:** This paper emanates from research supported by the ERA-NET Cofund grant under the CHIST-ERA IV Joint Call on Novel Computational Approaches for Environmental Sustainability (CES) project "Supporting Energy Communities- Operational Research and Energy Analytics" (SEC-OREA). Sandeep Araveti was funded by the Irish Research Council. The work of Cristian Aguayo was supported by the Fonds de la Recherche Scientifique-FNRS under Grant R801020F. The work of Anna Mutule and Evita Kairisa was supported by VIAA grant number ES RTD/2021/5. The work of Juan Pablo Sepulveda Adriazola was supported by ANR.

**Institutional Review Board Statement:** Not Applicable.

**Informed Consent Statement:** Not Applicable.

**Data Availability Statement:** ERA5 reanalysis data can be found here: https://cds.climate.copernicus.eu/cdsapp#!/dataset/reanalysis-era5-single-levels?tab=overview (accessed on 20 March 2021). MER-RA-2 reanalysis data can be found here: https://disc.gsfc.nasa.gov/datasets?project=MERRA-2 (accessed on 20 March 2021). The publicly available Met Éireann weather observation station wind speed data can be found here: https://www.met.ie/climate/available-data/historical-data (accessed on 20 March 2021). The mesoscale NEWA data can be accessed through the https://map.neweuropeanwindatlas.eu/. (accessed on 20 March 2021). The newer version of three PVGIS datasets can be found here: https://re.jrc.ec.europa.eu/pvg_tools/en/ (accessed on 1 February 2022).

**Acknowledgments:** This work emanates from research supported by the ERA-NET Cofund grant under the CHIST-ERA IV Joint Call on Novel Computational Approaches for Environmental Sustainability (CES) project "Supporting Energy Communities-Operational Research and Energy Analytics" (SEC-OREA). Sandeep Araveti was funded by the Irish Research Council.

**Conflicts of Interest:** The authors declare no conflict of interest. The funders had no role in the design of the study; in the collection, analyses, or interpretation of data; in the writing of the manuscript, or in the decision to publish the results.

**Abbreviations**

The following abbreviations are used in this manuscript:

| | |
|---|---|
| 3D VAR | Three-dimensional Variational Assimilation |
| 4D VAR | Four dimensional Variational Assimilation |
| DJF | December, January, February |
| ECMWF | European Centre for Medium Range Weather Forecasts |
| ERA | ECMWF Reanalysis |
| EU | European Union |
| GOES | Goddard Earth Observing System Model |
| IOA | Index of Agreement |
| IST | Irish Standard Time |
| JJA | June, July, August |
| JRC | Joint Research Community |
| LEC | Local Energy Community |
| LEM | Local Energy Model |
| MAM | March, April, May |
| ME | Mean Error |
| MERA | Met Éireann Reanalysis |
| MERRA | Modern Era Retrospective Reanalysis |
| NASA | National Aeronautics and Space Administration |
| NECP | National Energy and Climate Plan |
| NEWA | New European Wind Atlas |
| PVGIS | PhotoVoltaic Geographical Information System |
| R | Correlation Coefficient |
| REC | Renewable Energy Community |
| RES | Renewable Energy Source |
| RMSE | Root Mean Square Error |
| SON | September, October, November |
| WRF | Weather Research and Forecasting |
| WS | Wind Speed |

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
