# Peer review of "Wind Energy Assessment for Renewable Energy Communities"

_2674-032X, doi:10.3390/wind2020018_

Round 1
Reviewer 1 Report
General comments
The paper presents an interesting analysis of renewable energy communities with the aim of achieving consistent accuracy in the feasibility study.
The literature review is adequate and the references are well distributed across the text. The manuscript organization and the paragraph distribution are linear. The figures are well distributed but sometimes require a better explanation.
The paper includes an evaluation of the wind power potential based on the reliability of weather data between available datasets (sat-based) and measured data. The analysis deals with different information and evaluates the wind potential for a different location in Ireland.
The approach appears consistent and the conclusions are adequate.
Nevertheless, some aspects of the research need to be implemented. The main conclusion awards the ERA5 as the best available option for data sources. In my opinion, the difference between ERA5 and PV-GIS is not very impressive considering the unpredictability of the resource and the year-by-year variation of the conditions. Furthermore, the PV-GIS allows a better spatial resolution with a smaller grid that affects the discrepancy between the “real” location (measured data) and the dataset. However, the analysis of grid behavior differences exceeds the purpose of the paper and could be considered for future work.
In any case, the estimation of the wind potential is remarkable with respect to the standard design approach for renewable communities.
Concluding, the topic of the work appears clear and the results required minor improvements: in my opinion, after the required modification, the paper fulfills the Journal requirements.
Specific comments.
- Please add unit in Fig. 1.
- Please check typos.
- Please improve Fig. 5 graphic quality.
Author Response
We would like to thank the reviewer for their constructive response and feedback.
In response to the points raised:
Difference between ERA5 and PV-GIS is not very impressive: overall we decided in favour of ERA5 since on average across the metrics and locations ERA5 was marginally better on average, see tables 4 and 5. The IOA and R-squared are better for ERA5. We agree the analysis of grid behavior differences exceeds the purpose of the paper and could be considered for future work and have noted this in the revision.
Specific comments:
- Please add unit in Fig. 1.: we have updated the figure.
- Please check typos: we have rechecked the paper and corrected a small number of typos.
- Please improve Fig. 5 graphic quality: we have updated the figure with a higher quality version.
Reviewer 2 Report
Dear Authors,
Thank you for your contribution. I only have 2 minor changes to propose.
1) Do not use links on the main text. Instead convert them as references.
2) Some figures appear blurry. I understand that these may be produced from means where you cannot change the resolution. If possible download the raw data and generate your own figures using plotting software. Some blurry figures are Figure 4, Figure 8. Figure 11 needs larger fonts and better colouring of the lines because they fix up. You can you dashed lines etc.
Author Response
We thank the reviewer for the suggested minor changes:
1) Do not use links on the main text. Instead convert them as references: we have updated the paper as suggested and moved the links to the references.
2) Some figures appear blurry: we have rerun the figures to create higher resolution versions and improve the appearance.
Reviewer 3 Report
I would like to congratulate the authors on their work. Overall I think it’s a well written and well laid out paper. I believe it’s near publication ready but I would recommend the following:
- The abstract is quite brief. I would include more results of the study, similar to what’s in the conclusion.
- I would remove the banner from the top of Figure 1, SEAI could be referenced within the caption instead.
- How widely used is the method used in section 2.8? Has it been used for similar studies in the past with accuracy? This should be elaborated on further.
- The resolution of Figure 2 isn’t great. I would also change the colour scheme if possible and remove the title at the top.
Author Response
We thank the reviewer for the comments and suggestions. In response:
- The abstract is quite brief. I would include more results of the study, similar to what’s in the conclusion: this is a good suggestion, we have extended the abstract to include more of the conclusions.
- I would remove the banner from the top of Figure 1, SEAI could be referenced within the caption instead: we have amended the figure as recommended.
- How widely used is the method used in section 2.8? Has it been used for similar studies in the past with accuracy? This is a standard approach.
- The resolution of Figure 2 isn’t great: we have rerun the figure to improve their quality and appearance.